 **eLIFE**

# The basic leucine zipper transcription factor NFIL3 directs the development of a common innate lymphoid cell precursor

**Xiaofei Yu[1]\*[†], Yuhao Wang[1][†], Mi Deng[2,3], Yun Li[1], Kelly A Ruhn[1], Cheng Cheng Zhang[2,3], Lora V Hooper[1,4]\***

[1]Department of Immunology, University of Texas Southwestern Medical Center, Dallas, United States; [2]Department of Physiology, University of Texas Southwestern Medical Center, Dallas, United States; [3]Department of Developmental Biology, University of Texas Southwestern Medical Center, Dallas, United States; [4]Howard Hughes Medical Institute, University of Texas Southwestern Medical Center, Dallas, United States

**Abstract** Innate lymphoid cells (ILCs) are recently identified lymphocytes that limit infection and promote tissue repair at mucosal surfaces. However, the pathways underlying ILC development remain unclear. Here we show that the transcription factor NFIL3 directs the development of a committed bone marrow precursor that differentiates into all known ILC lineages. NFIL3 was required in the common lymphoid progenitor (CLP), and was essential for the differentiation of αLP, a bone marrow cell population that gives rise to all known ILC lineages. Clonal differentiation studies revealed that CXCR6+ cells within the αLP population differentiate into all ILC lineages but not T- and B-cells. We further show that NFIL3 governs ILC development by directly regulating expression of the transcription factor TOX. These findings establish that NFIL3 directs the differentiation of a committed ILC precursor that gives rise to all ILC lineages and provide insight into the defining role of NFIL3 in ILC development.

**\*For correspondence:** Xiaofei. Yu@utsouthwestern.edu (XY); lora.hooper@utsouthwestern.edu (LVH)

[†]These authors contributed equally to this work

**Competing interests:** The authors declare that no competing interests exist.

**Reviewing editor**: Satyajit Rath, National Institute of Immunology, India

## Introduction

Innate lymphoid cells (ILCs) are a recently identified family of lymphocytes that perform a variety of immune functions at barrier surfaces (*Spits and Cupedo, 2012*). Although ILCs share a common developmental origin with B- and T-cells, they lack antigen-specific receptors. Instead, they exert their immune functions through cytokine secretion in a manner similar to T helper cells (*Spits et al., 2013*). Despite their important contributions to immunity, the pathways that regulate ILC development remain poorly understood.

There are three known ILC groups. ILC1, which include conventional NK (cNK) cells, require the transcription factors T-BET and/or EOMES, produce interferon-γ (IFNγ) (*Kiessling et al., 1975*; *Gordon et al., 2012*; *Fuchs et al., 2013*), and promote immunity to intracellular pathogens (*Yokoyama et al., 2004*; *Klose et al., 2014*). ILC2 require the transcription factor GATA-3, produce IL-5/13 and amphiregulin (*Moro et al., 2010*; *Neill et al., 2010*; *Monticelli et al., 2011*; *Hoyler et al., 2012*), and promote tissue repair and anti-helminth immunity (*Monticelli et al., 2012*). ILC3, which include lymphoid tissue inducer (LTi) cells, depend on the transcription factor RORγt and secrete IL-17/22 (*Eberl and Littman, 2003*; *Satoh-Takayama et al., 2008*; *Luci et al., 2009*; *Takatori et al., 2009*). ILC3 are especially important for the defense of barrier surfaces as they promote anatomical containment of commensal bacteria (*Sonnenberg et al., 2012*), regulate CD4+ T cell responses to commensal bacteria (*Hepworth et al., 2013*; *Qiu et al., 2013*), and stimulate epithelial cells to produce antibacterial proteins (*Sanos et al., 2011*).

**eLife digest** The mucus-covered tissues that line the nose, mouth, and the digestive tract play an important role in protecting the body from infection. These mucosal tissues are the first line of defense against any pathogens we inhale or ingest, and help to keep communities of helpful bacteria—such as those that aid digestion—in place so that they can perform their beneficial functions without causing disease.

A special group of immune cells called innate lymphoid cells helps to prevent infection in the mucosal tissues and to repair damage to these tissues. There are several different types of innate lymphoid cells, with each type performing a different function. All innate lymphoid cells originate from precursor cells in the bone marrow. Some of these precursor cells had been identified previously, but were able to develop into only some of the different innate lymphoid cell types. Scientists suspected that a precursor cell existed that could develop into all types of innate lymphoid cell, but the identity of this cell had remained elusive.

Yu, Wang et al. now identify a precursor cell in the bone marrow that can produce all of the currently known different types of innate lymphoid cells. A protein called NFIL3 coaxes stem cells in the bone marrow into becoming these precursor cells, which only develop into innate lymphoid cells, and not into other immune cell types such as B cells and T cells.

Yu, Wang et al. find that NFIL3 causes some of these previously identified precursor cells to become dedicated producers of innate lymphoid cells by regulating another protein called TOX. Furthermore, gene therapy using NFIL3- or TOX-encoding DNA can help to restore normal numbers of innate lymphoid cells in mice whose bone marrow progenitor cells lack the NFIL3 gene.

These new details about how bone marrow stem cells develop into innate lymphoid cells may help scientists looking for new ways to treat infections or diseases that hamper the innate immune system.

All ILC differentiate from the common lymphoid progenitor (CLP), which resides in the bone marrow and also gives rise to B- and T-lymphocytes (*Possot et al., 2011*; *Hoyler et al., 2012*). Committed ILC progenitors that are positioned developmentally downstream of the CLP have been identified, and give rise to various ILC subsets. For example, an *Id2* (inhibitor of DNA binding 2)-expressing progenitor, known as the common 'helper-like' innate lymphoid progenitor (CHILP), gives rise to 'helper-like' ILC lineages including ILC2, ILC3 and a subgroup of ILC1 (*Klose et al., 2014*). PLZF-positive progenitors, termed ILCP, differentiate into non-NK ILC1, ILC2, and ILC3 (*Constantinides et al., 2014*). However, these progenitors do not differentiate into cNK cells (*Constantinides et al., 2014*; *Klose et al., 2014*), suggesting that a precursor that gives rise to all ILC subtypes remains to be identified.

NFIL3 (also known as E4BP4) is a basic leucine zipper transcription factor that controls a number of different immune processes, including cytokine expression (*Kashiwada et al., 2011*; *Kobayashi et al., 2011*; *Motomura et al., 2011*), IgE class switching (*Kashiwada et al., 2010*), and T$_H$17 cell differentiation (*Yu et al., 2013*). It was identified several years ago as an essential transcription factor in the differentiation of cNK cells (*Gascoyne et al., 2009*; *Kamizono et al., 2009*). More recently, NFIL3 has been shown also to be required for the development of non-NK ILC1 (*Klose et al., 2014*), ILC2 (*Geiger et al., 2014*; *Seillet et al., 2014a*), ILC3 (*Geiger et al., 2014*; *Klose et al., 2014*; *Kobayashi et al., 2014*; *Seillet et al., 2014a*), and LTi cells (*Geiger et al., 2014*; *Seillet et al., 2014a*). Thus, NFIL3 is essential for the development of all ILC lineages.

Here we show that NFIL3 is required for the development of a common ILC progenitor from the CLP. The progenitor population is marked by CXCR6, and resides in the $\alpha_4\beta_7^+$ $\alpha$LP bone marrow population, which can give rise to all ILC lineages. Clonal differentiation assays show that the CXCR6$^+$ precursors are committed ILC progenitors that differentiate into all ILC lineages but not B- or T-cells. Finally, we show that NFIL3 directs progenitor differentiation by directly regulating the expression of TOX, a known driver of ILC differentiation. These findings provide new insight into the defining role of NFIL3 in the differentiation of innate lymphoid cells.

## Results

### *Nfil3⁻/⁻* mice are deficient in bone marrow ILC progenitors downstream of the CLP

NFIL3 has recently been shown to be essential for the development of all ILC lineages (*Geiger et al., 2014*; *Seillet et al., 2014a*). Consistent with these findings, we observed that *Nfil3⁻/⁻* mice had lowered frequencies and absolute numbers of ILC2, ILC3 (including the NKp46⁺ subtype), cNK cells, and non-NK ILC1 (*Figure 1A*; *Figure 1—figure supplement 1*). *Nfil3⁻/⁻* mice also had fewer and smaller Peyer's patches in the small intestine and remaining Peyer's patches contained fewer LTi cells (RORγt⁺ LTβ⁺) than wild-type mice (*Figure 1—figure supplement 2*), indicating a deficiency in LTi cells that is consistent with the prior reports (*Geiger et al., 2014*; *Seillet et al., 2014a*). These data support the conclusion that NFIL3 is required for the development of all ILC lineages.

ILCs develop from common lymphoid progenitors (CLPs) in the bone marrow (*Possot et al., 2011*; *Hoyler et al., 2012*). To gain insight into the cellular origin of the broad ILC deficiency in *Nfil3⁻/⁻* mice, we first examined undifferentiated bone marrow precursors that were enriched by negative selection (*Figure 1—figure supplement 3*). In agreement with previous findings (*Male et al., 2014*; *Seillet et al., 2014b*), wild-type and *Nfil3⁻/⁻* littermates harbored similar frequencies of LSK cells (Lin⁻ Sca1⁺ cKit⁺) (*Figure 1—figure supplement 3*), which include hematopoietic stem cells (HSC) that give rise to all lymphoid and non-lymphoid hematopoietic cells. To test whether the requirement for NFIL3 was intrinsic to bone marrow precursors, we co-transferred wild-type and *Nfil3⁻/⁻* LSK cells into lethally irradiated mice and examined ILC subsets 5 weeks later. *Nfil3⁻/⁻* LSK cells generated fewer ILC2, ILC3, and NK1.1⁺ ILC3 in the small intestine, and fewer cNK and non-NK ILC1 in the liver (*Figure 1B*). These data indicate that the requirement for NFIL3 in ILC development is intrinsic to bone marrow progenitors.

Like LSK cells, CLP cells are also present in normal numbers in the bone marrow of *Nfil3⁻/⁻* mice (*Figure 2A*) (*Male et al., 2014*; *Seillet et al., 2014b*). To further determine whether the NFIL3 requirement was CLP intrinsic, we co-transferred wild-type and *Nfil3⁻/⁻* CLP cells into sublethally irradiated alymphoid *Rag2⁻/⁻;Il2rg⁻/⁻* mice. 5 weeks later, *Nfil3⁻/⁻* CLP had generated fewer ILC2, ILC3 in the small intestine and fewer CD90⁺ cNK and non-NK ILC1 in the liver (*Figure 1C*). (Because of the lower proliferation potential of CLP compared to LSK cells, progeny cells were fewer and we were not able to reliably enumerate NK1.1⁺ ILC3 following CLP co-transfer.) These findings reveal that the requirement for NFIL3 in ILC development is intrinsic to the CLP and are consistent with the findings of Seillet et al. (*Seillet et al., 2014a*).

The CLP gives rise to all lymphoid cells, including ILCs, T cells, and B cells. In contrast to ILC numbers, overall T and B cell numbers are not altered in *Nfil3⁻/⁻* mice (*Kashiwada et al., 2010*). The general requirement for NFIL3 in ILC development therefore suggested that NFIL3 might be essential for the development of ILC-committed precursors downstream of the CLP. To further investigate the cellular origin of the ILC developmental deficiency in *Nfil3⁻/⁻* mice, we analyzed various bone marrow precursor populations downstream of the CLP in wild-type and *Nfil3⁻/⁻* mice. *Nfil3⁻/⁻* mice had markedly fewer Flt3⁻ α₄β₇⁺ CLPs (known as αLPs) (*Figure 2A*; *Figure 2—figure supplement 1A*), which have been shown to differentiate into ILC3 and NK cells (*Possot et al., 2011*).

*Nfil3⁻/⁻* mice also had fewer previously identified precursor cells that have a more restricted differentiation potential. These cells include ILC2 progenitor cells (ILC2P, Lin⁻ α₄β₇⁺ CD127⁺ Sca1⁺ CD25⁺) that only differentiate into ILC2 (*Hoyler et al., 2012*) (*Figure 2B*; *Figure 2—figure supplement 1B*), and the CHILP that can give rise to non-NK ILC1, ILC2 and NK1.1⁺ NKp46⁺ ILC3 (*Figure 2B*; *Figure 2—figure supplement 1B*). Similarly, NFIL3 has been found to be critical for generation of the earliest NK-committed precursors (PreNKP) (*Male et al., 2014*; *Seillet et al., 2014b*). Thus, *Nfil3⁻/⁻* mice have reduced numbers of precursors that give rise to cNK cells, non-NK ILC1, ILC2 and ILC3. Together, these data indicate that NFIL3 is required for generation of ILC precursors in the bone marrow.

### αLP differentiate into all ILC lineages

It is thought that ILCs differentiate from a common ILC progenitor population (*Spits et al., 2013*; *Tanriver and Diefenbach, 2014*). Prior studies have identified progenitor populations that develop into most, but not all, subtypes of known ILC lineages (*Hoyler et al., 2012*; *Constantinides et al., 2014*; *Klose et al., 2014*). The CHILP, identified through *Id2* lineage tracing studies, can differentiate into non-NK ILC1, ILC2 and NK1.1⁺ NKp46⁺ ILC3 but not cNK cells (*Klose et al., 2014*), and the PLZF-dependent ILCP gives rise to all ILCs except cNK cells (*Constantinides et al., 2014*). These

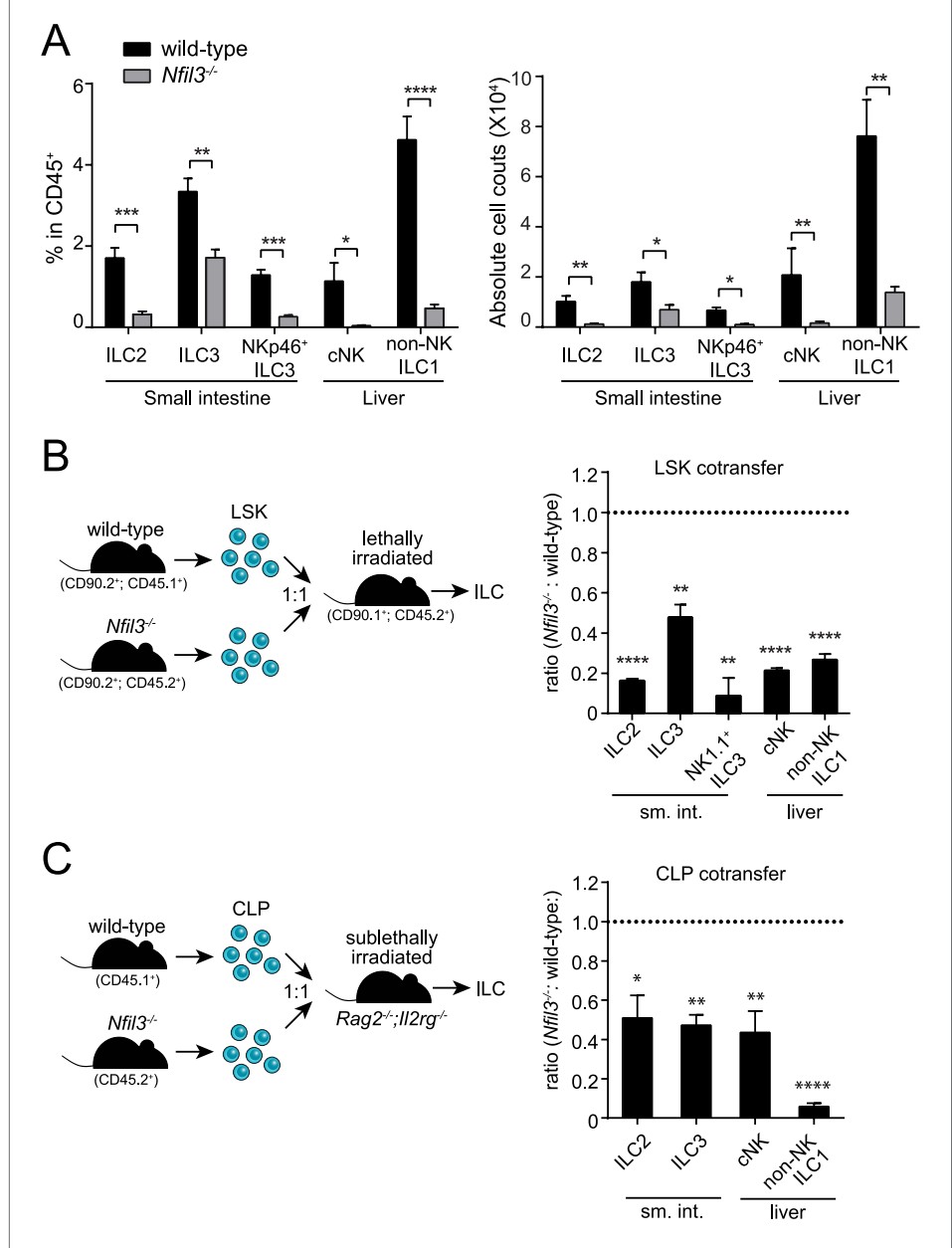

**Figure 1**. NFIL3 is required for innate lymphoid cell development in a cell-intrinsic manner. (**A**) *Nfil3⁻/⁻* mice show reduced frequencies (left panel) and numbers (right panel) of major ILC types, including conventional NK (cNK), non-NK ILC1, ILC2 and ILC3. Lymphocytes were isolated from the small intestinal lamina propria and the liver and were stained as described in Materials and methods. Gating strategies are depicted in *Figure 1—figure supplement 1*. cNK cells were identified as CD45⁺ Lin(CD3ε, CD19, CD5, TCRβ, TCRγδ)⁻ NK1.1⁺ T-BET⁺ EOMES⁺; non-NK ILC1 as CD45⁺ Lin(CD3ε, CD19, CD5, TCRβ, TCRγδ)⁻ NK1.1⁺ T-BET⁺ EOMES⁻; ILC2 as CD45⁺ Lin(CD3ε, CD19)⁻ GATA3⁺ Sca1⁺ KLRG1⁺; and ILC3 as CD45⁺ Lin(CD3ε, CD19)⁻ RORγt⁺ CD127⁺. The NK receptor-expressing subtype of ILC3 (also known as NK22 cells) was identified by additional staining for NKp46. (**B**) NFIL3 regulates ILC development in a bone-marrow progenitor intrinsic manner. Equal numbers of wild-type (CD90.2⁺ CD45.1⁺) and *Nfil3⁻/⁻* (CD90.2⁺ CD45.2⁺) LSK cells were co-transplanted into lethally irradiated CD90.1⁺ mice. Liver CD90⁺ NK and non-NK ILC1 and intestinal ILC2 and ILC3 were analyzed 4-6 weeks later. The ratios of ILCs derived from wild-type (CD45.1⁺) and *Nfil3⁻/⁻* (CD45.2⁺) donor cells were calculated and plotted. Significant variation from 1.0 is indicated by *. sm. int., small intestine. (**C**) *Nfil3* regulates ILC development in a CLP-intrinsic manner. Equal numbers of wild-type (CD45.1⁺) and *Nfil3⁻/⁻* (CD45.2⁺) CLPs were co-transplanted into sublethally irradiated alymphoid *Rag2⁻/⁻;Il2rg⁻/⁻* mice. ILCs were analyzed 4–6 weeks later as for the LSK experiment. Groups were compared by

*Figure 1. Continued on next page*

*Figure 1. Continued*

two-tailed student's t-test (**A**), one-sample t-test (**B**, LSK) or Wilcoxon signed rank test (**B**, CLP). Means ± SEM are shown. \*, p < 0.05, \*\*, p < 0.01, \*\*\*, p < 0.001, \*\*\*\*, p < 0.0001.
The following figure supplements are available for figure 1:
**Figure supplement 1**. Gating strategy for ILC analysis.
**Figure supplement 2**. *Nfil3*<sup>−/−</sup> mice are deficient in Peyer's patches and lymphoid tissue inducer cells.
**Figure supplement 3**. LSK cells are not deficient in *Nfil3*<sup>−/−</sup> mice.

findings accord with the partial ILC deficiencies seen in mice lacking *Id2* and *Zbtb16* (encoding PLZF) (*Boos et al., 2007*; *Savage et al., 2008*). In particular, cNK cell development is not impaired in *Zbtb16*<sup>−/−</sup> mice, while *Id2*<sup>−/−</sup> mice show cNK developmental defects only during NK maturation (*Boos et al., 2007*). Similarly, ILC2Ps are lineage-specified progenitors of ILC2s with no appreciable potential to differentiate into cNK cells or ILC3 (*Hoyler et al., 2012*). The broad ILC deficiency (including cNK cells) and impaired ILC precursor development in *Nfil3*<sup>−/−</sup> mice thus suggested that the NFIL3 might be required for the generation of a common ILC progenitor that lies developmentally upstream of the previously identified ILC precursors. We therefore sought to identify NFIL3-dependent precursor populations that differentiate into all ILC lineages.

In contrast to CHILP, ILCP, and ILC2P, αLP cells can differentiate into both cNK cells and ILC3 (*Yoshida et al., 2001*; *Possot et al., 2011*) and thus likely represent an earlier stage of ILC development. This idea is supported by expression profiles of key transcription factors known to be involved in ILC development (*Figure 2C*). Similar to CLP and CHILP, αLPs do not express transcription factors that specify ILC lineages, such as RORγt, GATA3, T-BET and EOMES, suggesting an undifferentiated phenotype. However, in contrast to CHILPs, which uniformly express high levels of ID2, only a small fraction of αLPs are ID2<sup>+</sup> (*Figure 2C*). The majority of αLPs express ID2 at levels that are markedly lower than those in CHILPs. Because ID2 is virtually undetectable in CLPs, this suggests that αLPs may represent a transitional stage between CLP and CHILP. This is further supported by the fact that αLPs do not express PLZF while a major fraction of CHILPs express PLZF (*Klose et al., 2014*), which defines another group of ILC precursors (ILCP) that lack cNK cell differentiation potential (*Constantinides et al., 2014*).

To determine whether αLP can also give rise to ILC2, we co-cultured purified αLP with bone marrow stromal OP9 cells (OP9-GFP) or OP9 cells expressing the Notch ligand Delta-like 1 (OP9-DL1), which support ILC differentiation in vitro (*Holmes and Zúñiga-Pflücker, 2009*; *Possot et al., 2011*; *Hoyler et al., 2012*). When co-cultured with OP9-DL1 cells in the presence of ILC2-inducing cytokines, αLPs readily developed into ILC2 as the majority of progeny cells expressed ILC2 markers (GATA3<sup>+</sup> Sca1<sup>+</sup>) (*Figure 3A*). When OP9-GFP cells (not expressing Notch ligand) were used in this assay, only a small fraction of progeny cells became ILC2 (*Figure 3A*), confirming that Notch signaling is important for ILC2 differentiation in vitro (*Wong et al., 2012*; *Yang et al., 2013*). In agreement with a prior study (*Possot et al., 2011*), αLP differentiated into ILC3 and RORγt<sup>−</sup> NK1.1<sup>+</sup> cells under ILC3-inducing conditions (*Figure 3A*).

To assess the potential of αLPs to differentiate into ILC2 in vivo, we transferred ~1000 purified αLPs (CD45.1<sup>+</sup>) into sublethally irradiated *Rag2*<sup>−/−</sup>;*Il2rg*<sup>−/−</sup> mice (CD45.2<sup>+</sup>). After 5 weeks, ILC2 that had differentiated from engrafted αLPs were detected in small intestine and colon of the recipient mice (*Figure 3B*). We noted that GATA3<sup>+</sup> ILC2 comprised a small fraction of CD127<sup>+</sup> ILCs in the small intestine but were the majority in the colon while RORγt<sup>+</sup> ILC3 showed the reverse tissue distribution pattern (*Figure 3B*). This suggests that tissue-specific microenvironment influences ILC development or recruitment. Consistent with the previously reported cNK cell differentiation potential of αLPs (*Possot et al., 2011*), donor cells gave rise to cNK cells in the liver, and also differentiated into non-NK ILC1 (*Figure 3C*). Differentiation of ILCs from αLPs was not caused by contamination of αLPs with CLPs, as no donor-derived B cells were detected in the spleen and small intestine of recipient mice (*Figure 3D*). This accords with the loss of B cell differentiation potential in αLPs (*Yoshida et al., 2001*; *Possot et al., 2011*). However, there were small numbers of donor-derived T cells, consistent with prior findings that

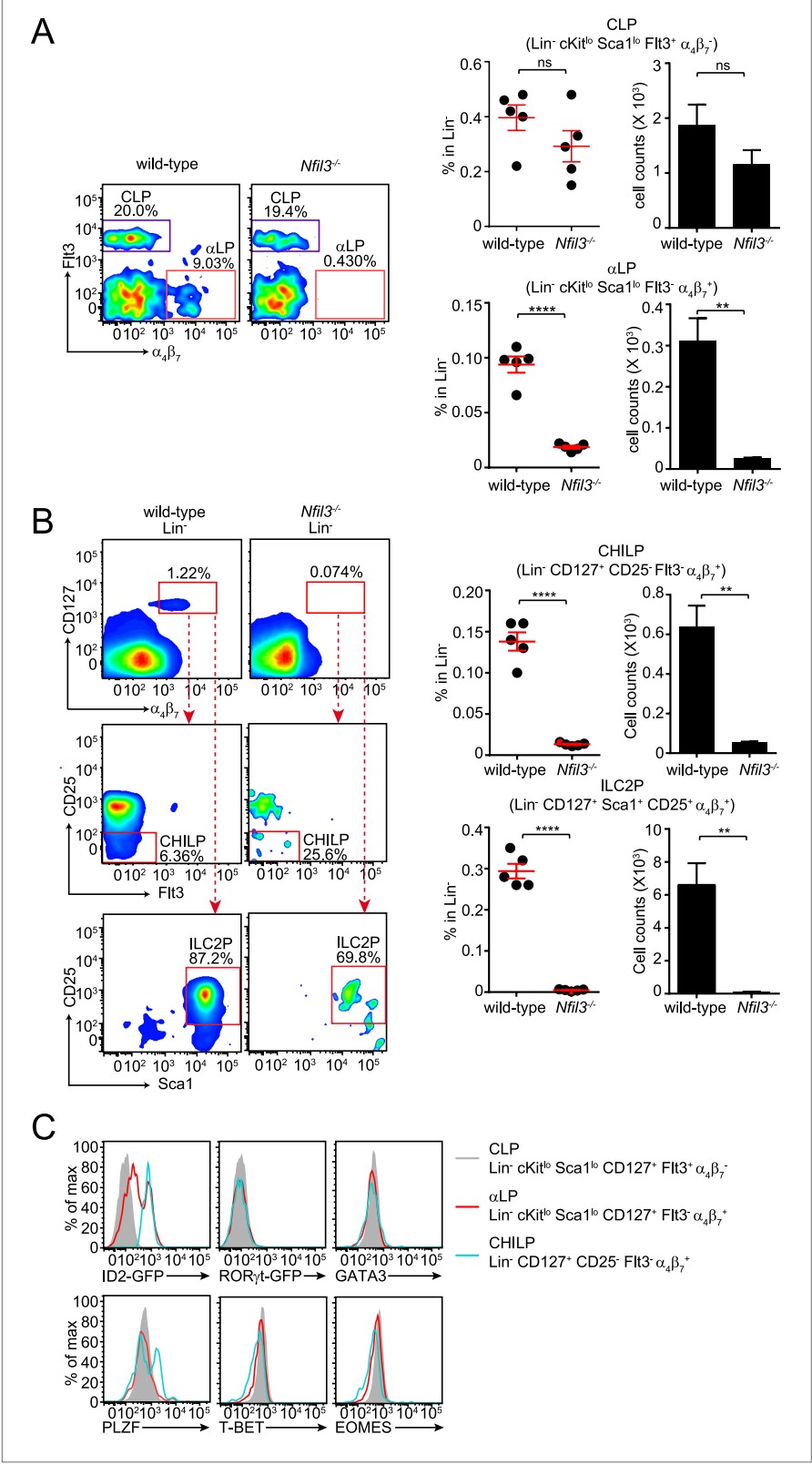

**Figure 2**. *Nfil3−/−* mice are deficient in bone marrow ILC precursors downstream of the CLP. (**A**) *Nfil3−/−* mice have comparable CLP frequencies but show deficiencies in αLP. Bone marrow cells were isolated from femur and tibia of wild-type and *Nfil3−/−* mice. Lineage marker (CD3ε, B220, CD11b, Gr-1, Ter-119, CD5, TCRγδ, NK1.1)-negative (Lin−)

*Figure 2. Continued on next page*

*Figure 2. Continued*

cells were first enriched by negative selection and then stained with antibodies to identify CLP (Lin⁻ cKit^low CD127⁺ Sca1^low Flt3⁺ α4β7⁻) and αLP (Lin⁻ cKit^low CD127⁺ Sca1^low Flt3⁻ α4β7⁺) (*Possot et al., 2011*). Gating strategy and representative flow plots are shown on the left and combined data for the frequencies and absolute numbers of CLP and αLP are shown on the right. (**B**) *Nfil3⁻/⁻* mice are deficient in common 'helper-like' innate lymphoid progenitor (CHILP) cells and ILC2P (*Klose et al., 2014*). Bone marrow cells were processed as above and CHILPs and ILC2Ps were identified as Lin⁻ CD127⁺ α4β7⁺ CD25⁻ Flt3⁻ and Lin⁻ CD127⁺ α4β7⁺ CD25⁺ Sca1⁺, respectively. (**C**) Expression of key transcription factors involved in ILC development in CLP, αLP and CHILP. Bone marrow cells were isolated from *Id2^GFP/+* and *Rorgt^GFP/+* mice to examine Id2 and RORγt expression. Expression of GATA3, PLZF, T-BET and EOMES were examined in C57BL/6 mice with specific antibodies. Statistical analysis was performed with two-tailed student's t-test. Means ± SEM are shown. ns, not significant; ***, p < 0.001, ****, p < 0.0001.

The following figure supplement is available for figure 2:

**Figure supplement 1**. Gating strategies for bone marrow lymphoid progenitor analysis.

---

αLP retain some T cell differentiation potential (*Possot et al., 2011*). Thus, αLPs can give rise to all known ILC lineages in vitro and in vivo. Given the more restricted differentiation potential of CHILP, ILCP, and ILC2P, αLPs are therefore likely to be developmentally upstream of these progenitors, and defective αLP development in *Nfil3⁻/⁻* mice can thus explain the general ILC deficiency in these mice.

## CXCR6⁺ αLP cells differentiate into all ILC lineages but not T- and B-cells

The residual T cell differentiation potential of αLPs suggested that this population includes cells that are not fully committed ILC precursors. Prior studies have shown that when αLP cells acquire CXCR6 expression, they continue to give rise to cNK cells and ILC3 but lose T cell differentiation potential (*Possot et al., 2011*). CXCR6⁺ cells comprised 3–4% of the αLP population in the bone marrows of adult wild-type and *Nfil3⁻/⁻* mice (*Figure 4A*, *Figure 4—figure supplement 1*). Their absolute numbers were diminished in *Nfil3⁻/⁻* mice (*Figure 4B*), in parallel with the decrease in total αLP numbers (*Figure 2A*). We therefore hypothesized that CXCR6⁺ αLPs might include NFIL3-dependent committed ILC precursors that give rise to all ILC lineages. To assess the developmental potential of CXCR6⁺ αLP cells, we isolated cells by flow cytometry and cultured individual cells with OP9-DL1 feeder cells in the presence of non-polarizing SCF and IL-7 (*Figure 4C,D*). In contrast to CXCR6⁻ αLP cells, which retained the ability to differentiate into T cells, CXCR6⁺ αLP cells failed to give rise to T cells in any of the clones examined (*Figure 4C*).

To determine if a single CXCR6⁺ αLP cell could give rise to all ILC lineages, we assessed the clonal differentiation potential of the CXCR6⁺ αLP cells in vitro. While no T cells were detected, mixtures of cNK, non-NK ILC1, ILC2, and ILC3 cells were present in the progeny populations of single CXCR6⁺ αLP cells (*Figure 4C,D*). Approximately 60% of wells with clonal growth contained multiple ILC lineages. In particular, 43.3% of the wells differentiated from individual CXCR6⁺ αLP cells contained two ILC lineages, and 11.7% three ILC lineages (*Figure 4C*). Importantly, 2.5% of wells contained all four ILC lineages, demonstrating that individual CXCR6⁺ αLP cells can differentiate into all known ILC lineages. In agreement with the in vitro differentiation data, CXCR6⁺ αLP cells differentiated into cNK, non-NK ILC1, ILC2 and ILC3, but not B cells or T cells, when transferred into sublethally-irradiated *Rag2⁻/⁻;Il2rg⁻/⁻* mice (*Figure 4E*). Thus, CXCR6⁺ αLP cells include committed ILC precursors that can differentiate into all major ILC lineages in vitro and in vivo.

## NFIL3-dependent ILC development is mediated by Tox

To identify potential mechanisms underlying NFIL3-dependent ILC development, we isolated CLPs from wild-type and *Nfil3⁻/⁻* mice and surveyed their transcriptomes by Illumina BeadArrays. *Nfil3* expression was readily detected in CLPs (*Figure 5A*), which accords with previous reports (*Geiger et al., 2014*; *Seillet et al., 2014b*) and is consistent with the finding that NFIL3 regulates ILC development in a CLP-intrinsic manner (*Figure 1C*; *Seillet et al., 2014a*). However, there was no detectable expression in CLPs of other transcription factors that are known to govern ILC development (*Figure 5A*). These factors include *Id2* (*Hoyler et al., 2012*; *Male et al., 2014*; *Seillet et al., 2014b*), *Zbtb16* (*Constantinides et al., 2014*), *Eomes* (*Male et al., 2014*; *Seillet et al., 2014b*), *Tcf7* (encoding TCF-1) (*Yang et al., 2013*), *Rora* (*Wong et al., 2012*), *Rorc* (*Eberl and Littman, 2003*; *Sawa et al., 2010*),

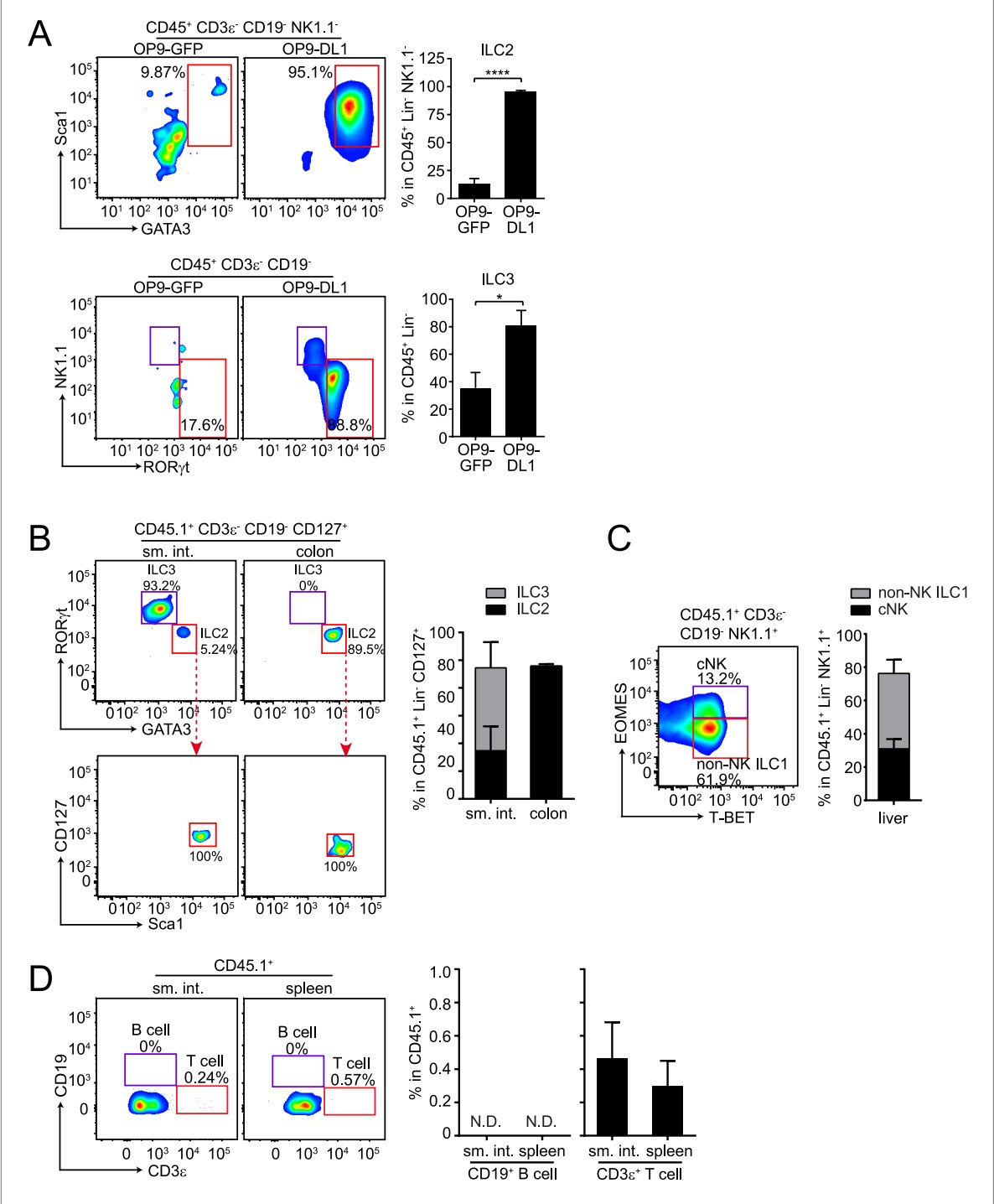

**Figure 3**. αLPs can differentiate into ILC2 in vitro and in vivo, and can thus give rise to all known ILC lineages. (**A**) αLPs can differentiate into ILC2 in vitro. αLPs were purified by FACS and ~25 cells were co-cultured with a bone marrow stromal cell line OP9 (OP9-GFP) or OP9 cells stably expressing the Notch ligand Delta-like 1 (OP9-DL1) for 14 days in the presence of ILC2-inducing (IL-2) or ILC3-inducing (IL-23) cytokines. Cells were then stained and analyzed by flow cytometry. ILC2 cells were identified as CD45+ CD3ε− CD19− GATA3+ Sca1+, ILC3 as RORγt+ NK1.1−, and ILC1 (including NK and non-NK ILC1) as CD45+ CD3ε− CD19− RORγt− NK1.1+. Typical flow plots are shown on the left and combined data are shown on the right. (**B** and **C**) αLPs can differentiate into ILC2, ILC3, cNK, and non-NK ILC1 in vivo. αLP cells were purified from wild-type (CD45.1+) mice and ~1000 αLP cells were transplanted into sublethally irradiated *Rag2−/−;Il2rg−/−* (CD45.2+) mice. ILCs in the small intestine and colon (**B**) or liver (**C**) were examined 4–6 weeks later. (**D**) αLPs failed to differentiate into B cells both in the small intestine and spleen. There were small numbers of T cells in both the small intestine and spleen. Statistical analysis was performed with two-tailed student's t-test. Means ± SEM are shown. N.D., not detected; *, p < 0.05; ***, p < 0.001.

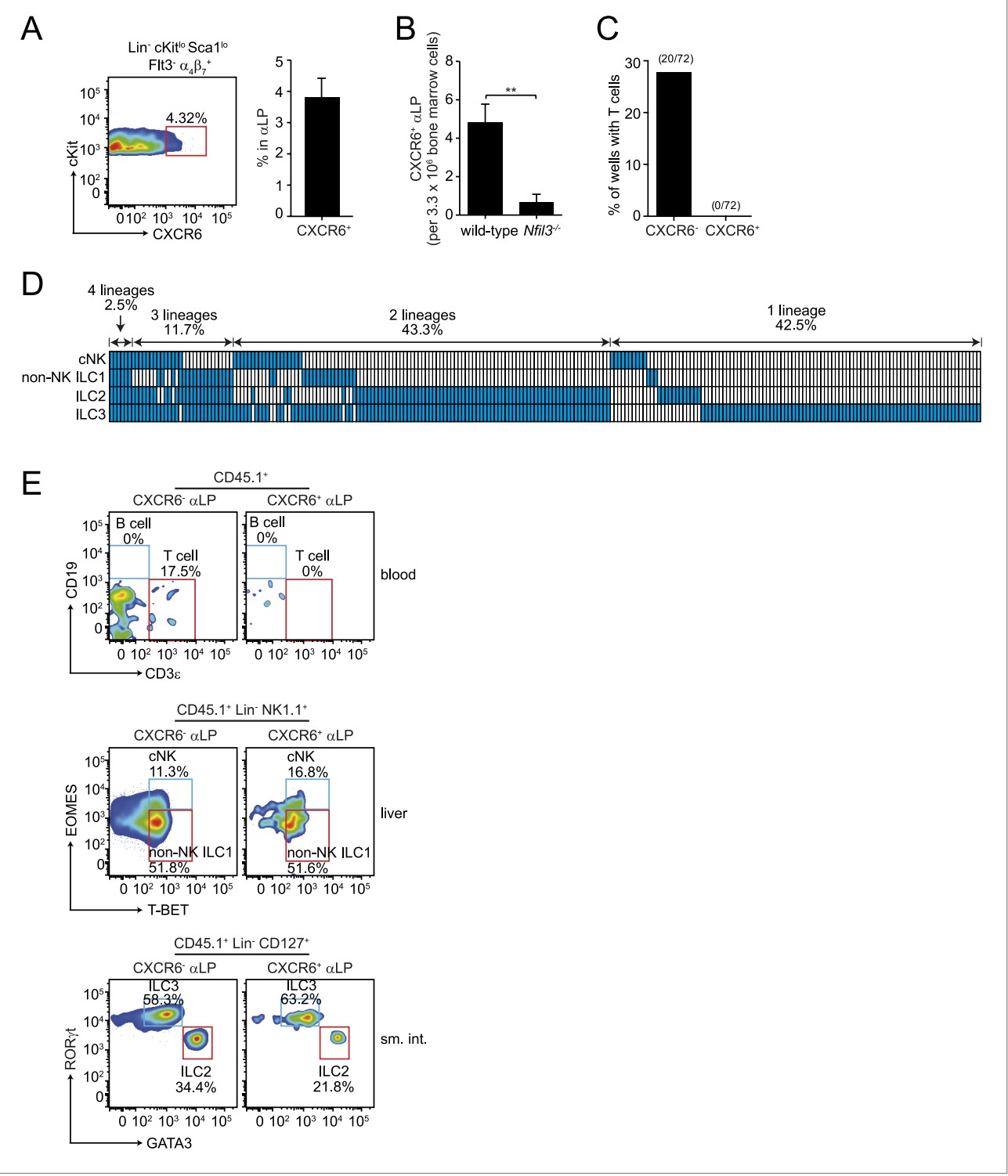

**Figure 4**. CXCR6+ αLP cells are ILC-committed precursors. (**A**) αLP can be divided into two subpopulations based on CXCR6 expression. (**B**) Enumeration of CXCR6+ cells in *Nfil3*−/− mice. Bone marrow progenitor cells were enriched by negative selection with Lin− cell counts ranging from 8–12.5 × 10⁶ per mouse. Lin− cells were then stained and analyzed by flow cytometry. The numbers of CXCR6+ αLP cells in 3.3 × 10⁶ Lin− cells are plotted. (**C**) CXCR6+ αLP cells lack T cell differentiation potential. CXCR6- and CXCR6+ αLP cells were individually sorted into the wells of a 96-well plate with an irradiated OP9-DL1 feeder cell monolayer. Cells were cultured in the presence of 20 ng/ml SCF and 20 ng/ml IL-7 for 3 weeks. T cells were detected by CD3ε staining. Data are shown as the percentages of CD45+ cell-containing wells in which T cells were detected. (**D**) CXCR6+ αLP cells are multipotent precursors to cNK

*Figure 4. Continued on next page*

*Figure 4. Continued*

cells, non-NK ILC1, ILC2 and ILC3 in vitro. Individual CXCR6$^+$ αLP cells were sorted and cultured as above (240 cells in total, pooled from two independent experiments). ILCs were analyzed by flow cytometry. cNK cells were detected as CD45$^+$ CD3ε$^-$ CD19$^-$ RORγt$^-$ GATA3$^-$ NK1.1$^+$ T-BET$^+$ EOMES$^+$; non-NK ILC1 as CD45$^+$ CD3ε$^-$ CD19$^-$ RORγt$^-$ GATA3$^-$ NK1.1$^+$ T-BET$^+$ EOMES$^-$, ILC2 as CD45$^+$ CD3ε$^-$ CD19$^-$ GATA3$^+$; and ILC3 as CD45$^+$ CD3ε$^-$ CD19$^-$ RORγt$^+$. Each well is presented as a column, with detected ILC lineages highlighted in blue. (**E**) CXCR6$^+$ αLP cells differentiate into cNK cells, non-NK ILC1, ILC2 and ILC3 in vivo. ~1000 CXCR6$^+$ αLP cells were purified from 20 CD45.1$^+$ mice by FACS sorting and were transplanted into sublethally irradiated *Rag2$^{-/-}$;Il2rg$^{-/-}$* (CD45.2$^+$) mice. T cells and B cells in the blood and ILCs in the small intestine and liver were examined 4–6 weeks later. Data shown are representative of two independent experiments. Statistical analysis was performed with two-tailed student's t-test. Means ± SEM are shown. **, p < 0.01.

The following figure supplement is available for figure 4:

**Figure supplement 1**. Frequencies of CXCR6$^+$ cells in wild-type and *Nfil3$^{-/-}$* αLP cell populations.

*Gata3* (**Hoyler et al., 2012**) and *Tbx21* (**Gordon et al., 2012**; **Rankin et al., 2013**). In contrast, the high mobility group (HMG) transcriptional regulator *Tox*, which is known to regulate NK and ILC3 development (**Aliahmad et al., 2010**), was expressed at a detectable level in wild-type CLPs and was down-regulated in *Nfil3$^{-/-}$* CLPs (**Figure 5A,B**). This suggested that NFIL3 might regulate *Tox* expression in CLPs.

CLPs are present in small numbers in adult mice (**Klose et al., 2014**), making it challenging to perform biochemical studies of *Tox* regulation by NFIL3 using these cells. As an alternative, we found that NFIL3 regulates *Tox* expression in EL4 cells, a mouse lymphoma cell line (**Figure 5C**). Knockdown of NFIL3 in EL4 cells with two independent shRNA constructs led to dose-dependent down-regulation of *Tox* expression (**Figure 5C**; **Figure 5—figure supplement 1**). Conversely, overexpression of NFIL3 in EL4 cells increased *Tox* expression (**Figure 5C**), indicating that *Tox* expression is sensitive to NFIL3 levels in EL4 cells in a manner similar to CLPs. A chromatin immunoprecipitation (ChIP) assay with an NFIL3-specific antibody (**Yu et al., 2013**) demonstrated that NFIL3 directly bound to the *Tox* promoter (nt −2105 to −1867) and that overexpression of NFIL3 enhanced this binding (**Figure 5D**). Finally, NFIL3 activated *Tox* promoter activity as assessed by a luciferase reporter assay (**Figure 5E**). Thus, NFIL3 activates *Tox* expression by directly binding to its promoter. EL4 cells are derived from T lymphocytes, a CLP-derived lineage, and thus we cannot exclude the possibility that the regulatory relationship between *Nfil3* and *Tox* differs between T lymphocytes and CLPs. Nevertheless, our studies on CLPs and EL4 cells both support the idea that NFIL3 is an activator of *Tox* expression.

Because *Tox* is known to be essential for cNK and ILC3 development (**Aliahmad et al., 2010**), we postulated that lowered *Tox* expression leads to the broad ILC deficiency in *Nfil3$^{-/-}$* mice and that restoring *Tox* expression would rescue ILC development. To test this idea, we cloned *Tox* coding sequences into a bicistronic vector (MSCV-IRES-hCD2), which allowed expression of the native form of TOX and also marked cells with the cell surface marker hCD2. We then delivered the TOX-encoding plasmid or the empty vector into purified *Nfil3$^{-/-}$* LSK cells (CD45.2$^+$) by retroviral transduction (**Zheng et al., 2012**; **Spencer et al., 2014**), followed by transfer of these cells into lethally irradiated wild-type mice (CD45.1$^+$) (**Figure 5—figure supplement 2**). Compared to the empty vector control, transduction of the TOX-encoding plasmid led to increased numbers of cNK cells in spleen, non-NK ILC1 cells in liver, and ILC2 and ILC3 in the small intestines of recipient mice (**Figure 5F**). We observed that rescue of ILC development from *Nfil3$^{-/-}$* LSK cells by *Tox* was largely comparable to rescue by *Nfil3* in the same setting, supporting the idea that *Tox* acts downstream of *Nfil3* in ILC development. Though ILC2 cells developing from *Tox*-rescued LSK cells were generally fewer than those from *Nfil3*-rescued LSK cells, the difference between the two groups was not statistically significant. Thus, ILC development is rescued by restoring *Tox* expression in *Nfil3$^{-/-}$* progenitors, indicating that NFIL3 drives ILC development in part by regulating *Tox* expression.

## NFIL3-dependent ILC development is essential for host defense against *Citrobacter rodentium* infection

IL-22 is produced both by ILC3 and T$_H$17 cells and is essential for protection against *Citrobacter rodentium* infection (**Satoh-Takayama et al., 2008**; **Zheng et al., 2008**). *Nfil3$^{-/-}$* mice show elevated susceptibility to intestinal pathogens such as *Citrobacter rodentium* (**Geiger et al., 2014**). However, *Nfil3$^{-/-}$* mice retain T$_H$17 cells, which are elevated relative to wild-type mice (**Yu et al., 2013**). To rule out confounding effects of T$_H$17 cells, we first crossed *Nfil3$^{-/-}$* mice with *Rag1$^{-/-}$* mice to create

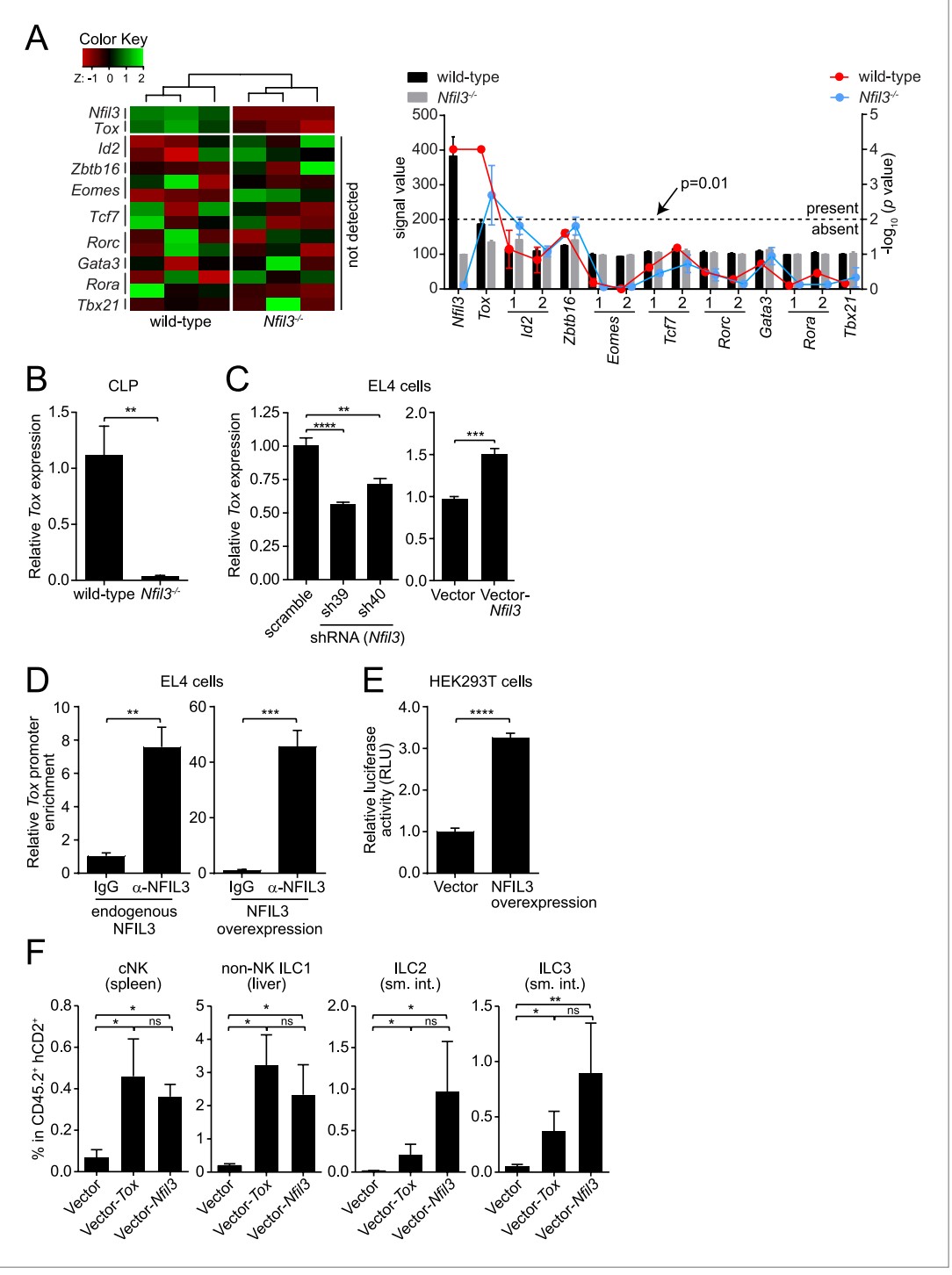

**Figure 5**. NFIL3-dependent ILC development is mediated by *Tox*. (**A** and **B**) *Tox* expression is lower in *Nfil3⁻/⁻* CLPs than in wild-type cells. (**A**) Heatmap comparing expression levels of transcription factors in wild-type and *Nfil3⁻/⁻* CLPs (left). Factors included *Nfil3*, *Tox* and other transcription factors that are known to be involved in ILC development. The absolute signal values and detection p values for each transcription factor in Illumina BeadArrays are also shown (right). Note that only *Nfil3* and *Tox* expression can be reliably detected in wild-type CLP cells. (**B**) Q-PCR analysis of *Tox* expression in wild-type and *Nfil3⁻/⁻* CLPs. (**C**–**E**) NFIL3 activates *Tox* expression by binding directly to the *Tox* promoter. (**C**) *Tox* expression was determined by Q-PCR following shRNA knockdown of NFIL3 (left), and NFIL3 overexpression (right) in EL4 cells. (**D**) ChIP analysis of EL4 cells using an NFIL3-specific antibody or IgG control. *Tox* promoter (nt −2105 to −1867) enrichment was calculated as the ratio of the
*Figure 5. Continued on next page*

*Figure 5. Continued*

NFIL3-specific antibody pull-down to the IgG control pull-down. The left panel shows results with endogenous NFIL3 levels and the right panel shows results with NFIL3 overexpression. (**E**) Luciferase reporter assay. A 2.8 kb fragment of the *Tox* promoter was cloned and fused with the firefly *luciferase* gene to generate a *Tox-luciferase* reporter. HEK293T cells were co-transfected with the reporter and an empty vector or an NFIL3-encoding vector. Luciferase activity was normalized to cells transfected with vector-only controls. (**F**) Restoring *Tox* expression in *Nfil3⁻/⁻* progenitors rescues ILC development in vivo. *Nfil3⁻/⁻* LSK cells (CD45.2⁺) were retrovirally transduced with either an empty vector (MSCV-IRES-hCD2), a TOX-encoding vector (MSCV-*Tox*-IRES-hCD2), or an NFIL3-encoding vector (MSCV-*Nfil3*-IRES-hCD2) and then transferred into lethally irradiated wild-type (CD45.1⁺) mice. ILCs were examined 5–6 weeks later. The frequencies of total ILC2 and ILC3 within CD45.2⁺ hCD2⁺ cells are shown. Statistical comparisons between groups were performed with two-tailed student's t-test (**B**–**E**), nonparametric one-way ANOVA test with posttests (**F**). Means ± SEM are shown. ns, not significant; *, $p < 0.05$; **, $p < 0.01$; ***, $p < 0.001$; ****, $p < 0.0001$.

The following figure supplements are available for figure 5:

**Figure supplement 1**. Knockdown of *Nfil3* by shRNA.

**Figure supplement 2**. Experimental design and gating strategy for the *Tox* rescue experiment.

---

*Nfil3⁻/⁻;Rag1⁻/⁻* mice, which lack T and B cells in addition to ILCs. *Nfil3⁻/⁻;Rag1⁻/⁻* mice were more susceptible to oral *C. rodentium* infection than *Rag1⁻/⁻* mice as measured by weight loss (**Figure 6**). These data thus suggest that NFIL3-dependent ILC development is essential for host immune defense against a mucosal pathogen.

## Discussion

Innate lymphoid cells are essential players in the immune response to various infections and in maintenance of barrier function in mucosal tissues. ILCs arise in the bone marrow from the CLP, and thus share a common developmental origin with T- and B-cells. The pathways that govern ILC differentiation downstream of the CLP have recently begun to be unraveled. Here we present new insight into the fundamental role of the basic leucine zipper transcription factor NFIL3 in ILC development. We show that NFIL3 directs the development of bone marrow precursors, derived from the CLP, which give rise to all known ILC lineages including cNK cells. Additionally, we show that NFIL3 regulates the expression of the transcription factor TOX, and provide evidence that a NFIL3-TOX transcription factor cascade is central to the development of all ILC lineages.

Several transcription factors are known to play essential roles in ILC development. For example, all ILC subsets express Id2, an antagonist of E proteins that control B- and T- cell commitment (**Kee, 2009**; **Hoyler et al., 2012**). Deletion of *Id2* in mice abrogates the development of multiple ILC lineages (**Yokota et al., 1999**; **Hoyler et al., 2012**), although NK developmental defects arise only during NK maturation (**Boos et al., 2007**). The HMG factor TOX is required for the development of cNK cells and ILC3 (**Aliahmad et al., 2010**), and the transcription factor PLZF is required for the differentiation of non-NK cell ILC subsets (**Constantinides et al., 2014**). Most recently, the basic leucine zipper transcription factor NFIL3 was

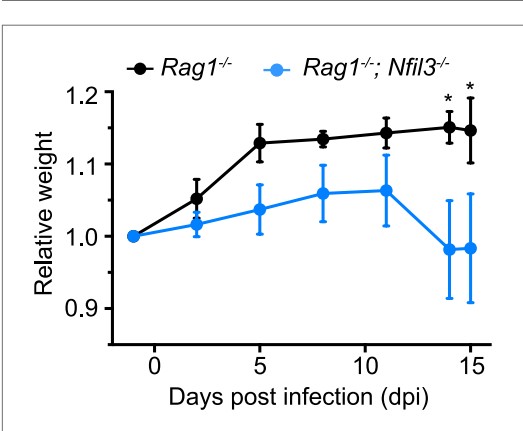

**Figure 6**. *Nfil3* deficiency results in increased susceptibility to *C. rodentium* infection in mice. *Nfil3⁻/⁻* mice were crossed with *Rag1⁻/⁻* mice to generate *Nfil3⁻/⁻;Rag1⁻/⁻* mice to eliminate the effects of adaptive immune cells, especially TH17 cells. *Rag1⁻/⁻* and *Nfil3⁻/⁻;Rag1⁻/⁻* mice were orally challenged with 5 × 10⁹ CFU of *C. rodentium* and mouse weight loss was monitored. 4 *Rag1⁻/⁻* mice and 5 *Nfil3⁻/⁻;Rag1⁻/⁻* mice were analyzed. Comparisons were carried out with or two-way ANOVA with posttests. Means ± SEM are shown. *, $p < 0.05$.

found to be essential for the development of all known ILC lineages, including cNK cells (*Geiger et al., 2014*; *Seillet et al., 2014a*).

Lineage tracing studies with *Id2* and *Zbtb16* (encoding PLZF) reporter mice have identified two distinct progenitor populations that develop into multiple ILC lineages but not cNK cells (*Hoyler et al., 2012*; *Constantinides et al., 2014*; *Klose et al., 2014*). The PLZF-dependent ILCP differentiates into non-NK ILC1, ILC2, and ILC3, but not cNK cells (*Constantinides et al., 2014*). Similarly, the common 'helper-like' innate lymphoid progenitor (CHILP) can differentiate into non-NK ILC1, ILC2 and NK1.1+ NKp46+ ILC3 but not cNK cells (*Klose et al., 2014*). This suggests that the ILCP and the CHILP may lie downstream of a common ILC progenitor that gives rise to all ILC lineages including cNK cells.

We have provided evidence that NFIL3 is required for the differentiation of a multipotent ILC precursor population, the αLP, from the CLP. αLP gave rise to all ILC lineages including cNK cells. Although αLP lack B cell differentiation potential, they retain some residual T cell differentiation potential (*Possot et al., 2011*). However, the CXCR6+ αLP subpopulation, which accounted for ~4% of αLP cells in adult bone marrow, differentiated into all ILC lineages including cNK cells, but not B- or T-cells. This suggests that CXCR6+ αLP represent committed ILC precursors that give rise to all ILC lineages including cNK cells. Thus, these cells are likely to lie developmentally upstream of the described ILCP and CHILP populations.

Our findings suggest that NFIL3 activation of TOX expression is a key mechanism by which NFIL3 directs ILC development. NFIL3 is required for *Tox* expression in the CLP, and directs *Tox* expression through direct binding to the *Tox* promoter. Since TOX has been shown to direct the development of multiple ILC lineages, including cNK cells and ILC3 (*Aliahmad et al., 2010*), this suggests that activation of TOX expression is a key mechanism by which NFIL3 influences ILC development. This idea is supported by our finding that forced *Tox* expression in *Nfil3*−/− bone marrow progenitors rescues the ILC developmental defect and restores differentiation of cNK, non-NK ILC1, ILC2, and ILC3. Together, these data support the idea that a NFIL3-TOX transcription factor cascade plays a fundamental role in the development of all ILC lineages. Recent studies have shown that forced expression of *Eomes* can rescue cNK cell development from *Nfil3*−/− hematopoietic progenitors (*Male et al., 2014*; *Seillet et al., 2014b*). However, because *Eomes* is not expressed in CLPs (*Figure 5A*; *Figure 5—figure supplement 1*) and *Eomes* deficiency only impacts cNK cells but no other ILCs (*Klose et al., 2014*), *Eomes* is unlikely to mediate the NFIL3-dependent development of non-NK ILCs, and may lie developmentally downstream of NFIL3-TOX during NK cell development.

Although NFIL3 is required for the development of the major ILC types and their precursors, some ILC subtypes appear to be NFIL3-independent. For example, certain NK cells, including salivary gland NK cells (*Cortez et al., 2014*) and tissue-resident NK cells (*Sojka et al., 2014*), are not impacted by *Nfil3* deficiency, in contrast to conventional NK cells. Extramedullary development of thymic NK cells is also independent of NFIL3 (*Crotta et al., 2014*). In addition, during mouse cytomegalovirus infection, NK cells in *Nfil3*−/− mice expand to numbers similar to those in wild-type mice through an IL-15-dependent mechanism (*Firth et al., 2013*). This suggests that the requirement for NFIL3 can be overridden by cytokine signaling during infection. Finally, despite the strict requirement for NFIL3 in bone marrow ILC precursor development (αLP and CHILP), *Nfil3*−/− mice appear to have normal lymph nodes (*Spits and Di Santo, 2011*; *Seillet et al., 2014b*) and only moderately impaired Peyer's patch development (*Figure 1—figure supplement 2*)(*Geiger et al., 2014*). One possibility is that fetal LTi cell function may be preserved in the absence of *Nfil3*, which could account for the presence of lymph nodes in *Nfil3*−/− mice.

*Nfil3* is regulated by the circadian clock and thus its expression varies diurnally in multiple tissues and cells (*Duez et al., 2008*; *Yu et al., 2013*). We previously showed that *Nfil3* expression varies diurnally in T cells and that NFIL3 synchronizes TH17 lineage specification to the day-night light cycle (*Yu et al., 2013*). Synchronization is essential for TH17 cell homeostasis, as circadian disruption by chronic light cycle perturbation elevates intestinal TH17 cell frequencies and increases susceptibility to intestinal inflammation (*Yu et al., 2013*). The finding that NFIL3 is required for the development of committed ILC precursors suggests that precursor differentiation may also be synchronized with diurnal light cycles through a similar mechanism. Future studies will examine whether *Nfil3* expression is diurnally regulated in the CLP, whether precursor generation is synchronized to circadian light cycles in an NFIL3-dependent manner, and whether disruption of circadian light cycles leads to dysregulated ILC development.

Altogether, our findings provide new insight into the defining role of NFIL3 in ILC development. Identification of a committed pan-ILC precursor should allow further insight into the developmental pathways that drive ILC cell fate decisions. Because of the general importance of ILCs in immune

defense, NFIL3-dependent pathways may provide new targets for treatment of inflammatory and infectious diseases.

## Materials and methods

### Mice

*Nfil3*$^{-/-}$ mice were obtained from Dr. Paul B. Rothman (Johns Hopkins University) (***Kashiwada et al., 2010***), and were maintained by heterozygous breeding in the Specific Pathogen Free (SPF) mouse facility at the University of Texas Southwestern Medical Center at Dallas. *Rag1*$^{-/-}$ mice (B6.129S7-Rag1$^{tm1Mom}$/J), CD90.1$^+$ mice (B6.PL-*Thy1*$^a$/CyJ), CD45.1$^+$ mice (B6.SJL-*Ptprc*$^a$ *Pepc*$^b$/BoyJ), Id2-eGFP reporter mice (B6.129S(Cg)-*Id2*$^{tm2.1Blh}$/ZhuJ), and RORγt-GFP reporter mice (B6.129P2(Cg)-*Rorc*$^{tm2Litt}$/J) were purchased from the Jackson Laboratory, Bar Harbor, Maine. *Nfil3*$^{-/-}$ mice were intercrossed with *Rag1*$^{-/-}$ mice to create *Nfil3*$^{-/-}$;*Rag1*$^{-/-}$ double knockout mice. *Rag2*$^{-/-}$;*Il2rg*$^{-/-}$ mice (B10;B6-*Rag2*$^{tm1Fwa}$ *Il2rg*$^{tm1Wjl}$) were purchased from Taconic Farms, New York. All procedures described in this study were performed in accordance with protocols approved by the Institutional Animal Care and Use Committees (IACUC) of the UT Southwestern Medical Center.

### Isolation and analysis of intestinal lamina propria lymphocytes

Lamina propria lymphocytes (LPLs) were isolated from the intestine as previously described (***Yu et al., 2013***). Briefly, intestines were dissected from mice and Peyer's patches were removed. Intestines were cut into small pieces and thoroughly washed with ice-cold PBS. Epithelial cells were removed by incubating intestinal tissues in Hank's buffered salt solution (HBSS) supplemented with EDTA and DTT, followed by extensive washing with PBS. Residual tissues were digested by Collagenase IV (Sigma, St. Louis, Missouri), DNase I (Sigma) and Dispase (BD Biosciences, San Jose, California) for 1 hr at 37°C. Cells were filtered through 100 μm cell strainers and applied onto a 40%:80% Percoll gradient (GE Healthcare, Pittsburgh, Pennsylvania), in which lamina propria lymphocytes were found at the interface of 40% and 80% fractions.

Livers were dissected from mice and cut into small pieces, followed by digestion with Collagenase IV (Sigma), DNase I (Sigma) and Dispase (BD Biosciences) for 1 hr at 37°C. Residual tissues were forced through 100 μm cell strainers. Cells were spun down and applied onto a 40%:80% Percoll gradient as for the LPLs.

Isolated lymphocytes were washed with PBS with 2 mM EDTA and 3% fetal bovine serum (FBS) and F$_c$ receptors were blocked with α-CD16/32 (2.4G2). Cells were then stained with antibodies against cell surface markers including α-CD3ε (500A2), α-CD19 (ebio1D3), α-CD5 (53-7.3), α-TCRβ (H57-597), α-TCRγδ (GL3), α-NK1.1 (PK136), α-Sca1 (D7), α-KLRG1 (2F1), α-NKp46 (29A1.4), α-CD45 (30-F11), α-CD45.1 (A20), α-CD45.2 (104), α-hCD2 (RPA-2.10), and α-CD127 (A7R34). Cells were fixed/permeabilized with eBiosciences (San Diego, California) Mouse Regulatory T Cell Staining Kit #3 per the manufacturer's instructions, and subjected to nuclear staining with α-RORγ (AFKJS-9), α-GATA3 (TWAJ), α-T-BET (4B10) and α-EOMES (Dan11mag). Cells were analyzed with an LSRII (BD Biosciences, San Jose, California) or CyAn ADP (Beckman Coulter, Jersey City, New Jersey) flow cytometer and data were processed with FlowJo software (Tree Star, Ashland, Oregon).

### Isolation and analysis of bone marrow progenitors

Femur and tibia were dissected from adult mice and bone marrow cells were released in PBS buffer containing 2 mM EDTA and 3% FBS with a mortar and pestle. Cells were filtered through 70 μm cell strainers and blocked with α-CD16/32 (2.4G2), followed by incubation with biotinylated lineage markers (Lin) antibodies including α-CD3ε (145-2C11), α-B220 (RA3-6B2), α-CD11b (M1/70), α-Gr1 (RB6-8C5), α-Erythroid Cells (TER119), α-CD5 (53-7.3), α-TCRγδ (GL3), and α-NK1.1 (PK136). Cells were then washed and incubated with α-biotin magnetic microbeads (Miltenyi Biotec, San Diego, California). Lineage-negative cells were enriched by an autoMACS sorter with the 'Depletes' setting. Surface staining was performed with antibodies including α-biotin (Bio3-18E7), α-CD45 (30-F11), α-cKit (2B8), α-CD127 (A7R34), α-Sca1 (D7), α-Flt3 (A2F10) and α-α$_4$β$_7$ integrin (DATK32) and α-CXCR6 (221002). FACS sorting was performed with a FACSAria cell sorter (BD Biosciences) while flow cytometry analysis was carried out with an LSRII (BD Biosciences). In both cases, LSK cells were identified as Lin$^-$ Sca1$^+$ cKit$^+$, CLP as Lin$^-$ cKit$^{low}$ CD127$^+$ Sca1$^{low}$ Flt3$^+$ α$_4$β$_7^-$, αLP as Lin$^-$ cKit$^{low}$ CD127$^+$ Sca1$^{low}$ Flt3$^-$ α$_4$β$_7^+$, ILC2P as Lin$^-$ CD127$^+$ α$_4$β$_7^+$ CD25$^+$ Sca1$^+$, and CHILPs as Lin$^-$ CD127$^+$ α$_4$β$_7^+$ CD25$^-$ Flt3$^-$. Data were processed with FlowJo software (Tree Star).

## Cell transfer assay

αLPs were purified from CD45.1⁺ mice by FACS sorting as described above. In order to obtain a large number (~1000) CXCR6⁺ αLP, femur and tibia from 20 CD45.1⁺ mice were pooled together for the cell isolation. *Rag2⁻/⁻;Il2rg⁻/⁻* recipient mice were sublethally irradiated with a dose of 4.2 Gy on the same day with an XRAD320 irradiator (Precision X-ray, Inc, North Branford, Connecticut). Cells were transplanted into recipient mice by retro-orbital injection. ILCs in recipient mice were examined 4–6 weeks later.

For CLP co–transfer experiments, wild-type and *Nfil3⁻/⁻* CLP cells were purified by FACS sorting and mixed at a 1:1 ratio before transplantation into sublethally irradiated *Rag2⁻/⁻;Il2rg⁻/⁻* recipient mice.

For LSK co-transfer, CD90.1⁺ recipient mice were lethally irradiated with two doses of 5 Gy on the same day. FACS-purified LSK cells from wild-type and *Nfil3⁻/⁻* mice were mixed at a 1:1 ratio and transplanted into recipient mice by retro-orbital injection.

## In vitro differentiation assays

αLPs were purified by FACS sorting as described above. For bulk culture, ~25 cells were co-cultured on a monolayer of OP9 cells (OP9-GFP) or OP9 cells stably expressing the Notch ligand Delta-like 1 (OP9-DL1) in αMEM media. To induce ILC2, the culture medium was supplemented with 20 ng/ml Stem Cell Factor (SCF, PeproTech, Rocky Hill, New Jersey), 20 ng/ml IL-7 (BioLegend, San Diego, California) and 20 ng/ml IL-2 (BioLegend). To induce ILC3, 20 ng/ml SCF, IL-7 and IL-23 (BioLegend) were added to the medium. The culture medium was replaced every 3–4 days and, after 14 days, cells were stained and analyzed by flow cytometry. For clonal differentiation, OP9-DL1 cells were irradiated at 1500 rad and seeded at a density of 10,000 cells per well in a 96-well plate. On the following day, CXCR6⁺ αLP cells were individually sorted into the wells and cultured in αMEM media supplemented with 20 ng/ml SCF and IL-7. Cells were analyzed by flow cytometry 3 weeks later. In total, 240 cells from two independent experiments were analyzed.

## CLP transcriptome analysis

CLPs from wild-type and *Nfil3⁻/⁻* mice were purified as described above. Total RNA was isolated with the PicoPure RNA Isolation Kit (Life Technologies, Grand Island, New York). RNA quality and quantity were determined with a Bioanalyzer (Agilent Genomics) with a Pico chip. Samples with RNA integrity numbers (RIN) larger than 8 were subjected to further processing and hybridized to the Mouse WG-6 V2 BeadChips (Illumina, San Diego, California) by the UT Southwestern Microarray Core Facility.

Microarray images were processed and annotated with GenomeStudio (Illumina). Differential gene expression analysis was performed using R together with bioConductor and the Limma package (*Smyth et al., 2005*; *Ritchie et al., 2011*). Briefly, signal intensities were first $\log_2$-transformed, followed by background correction and quantile normalization with the NEQC function. Empirical reliabilities of samples were estimated by the arrayWeights function, which gave each sample a weight score accordingly. Samples were then fitted into a weighted linear model by lmFit to detect differentially expressed genes.

## Tox rescue experiment

TOX-coding or NFIL3-coding sequences (CDS) were cloned by PCR from total mouse thymus cDNA into the bicistronic retroviral vector MSCV-IRES-hCD2 (a gift from Dr Chandrashekhar Pasare at UT Southwestern) to generate a TOX-encoding plasmid, MSCV-*Tox*-IRES-hCD2 and an NFIL3-encoding plasmid, MSCV-*Nfil3*-IRES-hCD2. The MSCV-IRES-hCD2, MSCV-*Tox*-IRES-hCD2, and MSCV-*Nfil3*-IRES-hCD2 plasmids were transfected into the Plat-E packaging cell line (*Morita et al., 2000*) with FugeneHD (Promega, Madison, Wisconsin) to produce retroviral particles. Cell culture supernatant was harvested 48 and 72 hr post transfection. Cell debris was first cleared by spinning at 400×*g* for 10 min, followed by passage through 0.2 µm sterile filters.

LSK cells were purified from *Nfil3⁻/⁻* mice by FACS sorting as described above and seeded into round-bottom 96-well plates at a density of 10,000 cells/well in STEMSPAN Serum-Free Expansion Medium (SFEM) (Stemcell Technologies, Vancouver, Canada) (*Zheng et al., 2012*). During retroviral transduction, cells were mixed with an equal volume of cleared retrovirus-containing cell culture supernatant, supplemented with 2 U/ml Heparin (Sigma), 10 ng/ml mouse Stem Cell Factor (SCF, Peprotech), 20 ng/ml mouse Thrombopoietin (TPO, Peprotech), 10 ng/ml mouse Fibroblast Growth Factor (FGF-1, Life Technologies) and 4 µg/ml polybrene (Sigma). Spinoculation was carried out at 1200×*g* for 90 min at 32°C to enhance retroviral transduction. 3 hr later, cell media was replaced with fresh STEMSPAN

SFEM media supplemented with the above cytokines but without polybrene. Transduction was performed on two consecutive days using retroviral supernatant harvested 48 and 72 hr post transfection, respectively.

On day 3, CD45.1+ wild-type recipient mice were lethally irradiated at two doses of 5 Gy as described above. LSK cells were collected from the 96-well plate with Cell Dissociation Buffer (Life Technologies) and washed with sterile PBS. 2000–4000 cells were transferred into recipient mice in 200 µl sterile PBS by retro-orbital injection. ILCs in recipient mice were examined 5–6 weeks later.

## shRNA knockdown of NFIL3

Five independent shRNA constructs (sh38-sh42) targeting mouse NFIL3 and a control construct containing scramble sequences (pLKO.1-scramble) were purchased from Sigma. To identify shRNA constructs that could effectively knock down NFIL3, 1 µg of shRNA plasmid and 1 µg of NFIL3-encoding plasmid (*Yu et al., 2013*) were co-transfected into HEK293T cells in a 6-well plate with FugeneHD (Promega). Cells were harvested 36 hr later, lysed and used for western blotting with anti-NFIL3 antibody.

Two shRNA constructs that could effectively knock down NFIL3 were identified: sh39 and sh40. These constructs as well as the pLKO.1-scramble vector were each co-transfected with the packaging plasmids pCMVDR9 and pVSVG into HEK293T cells. Cell culture supernatants were harvested 48 and 72 hr later and cleared by spinning and filtering as described above. Lentiviral particles were concentrated by ultracentrifugation at 75,000×*g* for 2 hr and resuspended in RPMI media.

EL4 cells were mixed with lentiviral particles in the presence of 4 µg/ml polybrene, and spinoculated at 1200×*g* for 90 min at 32°C. 2 days later, EL4 were selected with 8 µg/ml puromycin for 2 weeks. Live cells were sorted with a FACSAria cell sorter as they excluded propidium iodide.

## NFIL3 overexpression in EL4 cells

NFIL3 coding sequences were subcloned into MSCV-IRES-hCD2 to generate MSCV-*Nfil3*-IRES-hCD2. MSCV-IRES-hCD2 and MSCV-*Nfil3*-IRES-hCD2 were then transfected into Plat-E cells to produce retroviral particles as described above, which were then used to transduce EL4 cells. 3 days after transduction, EL4 cells were stained with anti-hCD2 and hCD2+ EL4 cells were purified with a FACSAria cell sorter. Sorted cells were maintained in RPMI media for another 3–4 days, followed by staining and sorting again. The resulting cells were expanded and *Nfil3* expression was examined by SYBR green-based real-time PCR.

## Chromatin Immunoprecipitation (ChIP)

ChIP experiments were carried out as previously described (*Yu et al., 2013*). Briefly, EL4 cells or NFIL3-overexpressing EL4 cells were cultured in RPMI medium at ~0.8 × 10⁶ cells/ml. Cells were harvested and fixed with 1% formaldehyde for 10 min in the dark, which was quenched by adding glycine to a final concentration of 0.15 M. Nuclei were released with a Dounce homogenizer (Wheaton, Millville, New Jersey) in Nuclear Isolation Solution containing 10 mM Tris pH 7.4, 5 mM MgCl$_2$, 25 mM KCl and 250 mM sucrose, and purified by spinning at 1000×*g* for 10 min over Hypertonic Solution containing 10 mM Tris pH 7.4, 5 mM MgCl$_2$, 25 mM KCl and 30% (wt/vol) sucrose. Purified nuclei were used for ChIP with the Magna ChIP assay kit (Millipore, Billerica, Massachusetts) per the manufacturer's instructions. The *Tox* promoter was detected by SYBR green-based real-time PCR with specific primers: Tox-ChIPF6: 5′-GACACTGACAGCAAGGACCA-3′ and Tox-ChIPR6: 5′-CAGGGCTTCATAGCACCGAT-3′, targeting nucleotide −2105 to nucleotide −1867 in the *Tox* promoter. Enrichment of the *Tox* promoter was determined by normalizing the level of the *Tox* promoter in DNA pulled down with an anti-NFIL3 antibody to that pulled down with an IgG control.

## *Tox-luciferase* reporter assay

A 2.3 kb fragment (−2133 to 232) of the *Tox* promoter was cloned into the pGL3-Basic vector to drive firefly *luciferase* expression (the *Tox-luciferase* reporter). HEK293T cells were cultured in a 96-well plate overnight and were co-transfected with the *Tox-luciferase* reporter and an empty or NFIL3-encoding vector. A pCMV-Renilla-Luciferase reporter was co-transfected into HEK293T cells to serve as an internal control. Luciferase activities were detected using the Dual-Glo Luciferase Assay kit (Promega) and measured with a SpectraMax M5e plate reader (Molecular Devices, Sunnyvale, California). Firefly luciferase activities in each sample were first normalized against Renilla luciferase activities in the same sample and then normalized against that in cells transfected with the empty vector.

## Citrobacter rodentium infection

The *C. rodentium* (DBS100) strain was originally obtained from ATCC (Manassas, Virginia). To infect mice, *C. rodentium* (DBS100) was first inoculated into Luria-Bertani (LB) broth overnight at 37°C with shaking in the presence of 50 μg/ml nalidixic acid, and was subcultured into fresh LB media the next morning until $OD_{600}$ = ~0.8–1.0. Bacteria were then harvested by centrifugation and resuspended in sterile PBS. *Rag1*$^{-/-}$ and *Nfil3*$^{-/-}$;*Rag1*$^{-/-}$ mice were deprived of food the night before infection and were orally gavaged with 5 × 10$^9$ CFU in 200 μl sterile PBS. The number of viable *C. rodentium* (DBS100) in the inoculum was confirmed by retrospective plating on nalidixic acid-containing LB-agar plates. Mouse disease conditions were monitored by weight loss.

## Acknowledgements

We thank Cassie Behrendt Boyd and Tess Leal for assistance with mouse experiments. We thank Dr Nicolai Van Oers and Ashley Hoover for sharing mouse thymus total cDNA, and Dr Hua Wang for assistance with cell irradiation. We also thank Dr Juan Carlos Zúñiga-Pflücker for providing us with the OP9 cell lines, and Dr Paul Rothman for providing the *Nfil3*$^{-/-}$ mice. This work was supported by NIH R01 DK070855 (LVH), a Burroughs Wellcome Foundation New Investigators in the Pathogenesis of Infectious Diseases Award (LVH), and the Howard Hughes Medical Institute (LVH).

## Additional information

### Funding

| Funder | Grant reference number | Author |
|---|---|---|
| National Institutes of Health | DK070855 | Lora V Hooper |
| Burroughs Wellcome Fund | New Investigators in the Pathogenesis of Infectious Diseases Award | Lora V Hooper |
| Howard Hughes Medical Institute | | Lora V Hooper |

The funders had no role in study design, data collection and interpretation, or the decision to submit the work for publication.

### Author contributions

XY, YW, CCZ, Conception and design, Acquisition of data, Analysis and interpretation of data, Drafting or revising the article; MD, YL, Conception and design, Acquisition of data, Analysis and interpretation of data; KAR, Acquisition of data, Analysis and interpretation of data; LVH, Conception and design, Analysis and interpretation of data, Drafting or revising the article

### Ethics

Animal experimentation: All animal experiments were approved by the Institutional Animal Care and Research Advisory Committee at the University of Texas Southwestern Medical Center, and the approved animal protocol number is 1004-06-04-1. The institutional guidelines for the care and use of laboratory animals were followed.

## Additional files

### Major dataset

The following dataset was generated:

| Author(s) | Year | Dataset title | Dataset ID and/or URL | Database, license, and accessibility information |
|---|---|---|---|---|
| Yu X, Wang Y, Deng M, Li Y, Ruhn KA, Zhang CC, Hooper LV | 2014 | The basic leucine zipper transcription factor NFIL3 directs the development of a common innate lymphoid cell precursor | GSE62337; http://www.ncbi.nlm.nih.gov/geo/query/acc.cgi?acc=GSE62337 | Publicly available at GEO (http://www.ncbi.nlm.nih.gov/geo/). |

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
