## [Decision Letter]

Thank you for sending your work entitled “NFIL3 directs the development of a common innate lymphoid cell precursor” for consideration at *eLife*. Your article has been favorably evaluated by Tadatsugu Taniguchi (Senior editor), a Reviewing editor, and 3 reviewers.

The Reviewing editor and the other reviewers discussed their comments before we reached this decision, and the Reviewing editor has assembled the following comments to help you prepare a revised submission.

The manuscript presents a substantial body of data supporting the idea that the transcription factor NFIL3 is crucial for the entire innate lymphoid cell (ILC) lineage commitment by showing, inter alia, that NFIL3 deficiency results in a strong reduction of the 'alpha-LP' lymphoid progenitor (aLP) cells that contain precursors for cNK and ILC3 cells, as well as a loss of the putative downstream common 'helper-like' innate progenitor (CHILP) cells. Further, the manuscript significantly extends earlier observations by showing that the ILC progenitor potential in aLP compartment is likely to be contained in the previously identified CXCR6-expressing sub-population. Finally, using a transcriptome survey, the manuscript generates the hypothesis that Tox is a downstream target of Nfil3 relevant for ILC development, and tests it with Tox-specific siRNA and overexpression approaches to correlate Nfil3 expression with Tox, to show that TOX is a downstream target of NFIL3, and to demonstrate that it can rescue ILC development of NFIL3-/- precursors.

While other groups have recently published similar findings, this manuscript makes important extensions in the cellular-molecular understanding of ILC development. The data are of high quality and rapid publication is warranted. However, some issues of concern as detailed below should be resolved prior to publication.

1) Since Nfil3-deficient mice appear to have lymph nodes (e.g., [47]) and Peyer's patches albeit fewer and smaller (present manuscript), it is arguable that LTi cell function is largely preserved during the fetal period in them. Also, NKp46- ILC3 are only mildly reduced in Nfil3-deficient mice (Figure 1). On the other hand, the authors show convincing data for the deficiency of the CHILP compartment, shown to be a common progenitor for helper-like ILCs including LTi cells, in the absence of Nfil3. Is it possible, for example, that Nfil3 is required for the maintenance of ILC progenitors in adult mice but not as strictly required for ILC fate decisions in the fetus? There is also literature suggesting Nfil3-indepedent generation of some ILC subsets; salivary gland NK cells (4), cNK cells after MCMV infection (7), as also the mild reduction in NKP cells in Nfil3-deficient mice (24). The discussion would benefit from acknowledgment and attempted resolution of these issues.

2) A fair proportion of the molecular connections between Nfil3 and Tox have been derived from studies with the EL4 lymphoma cell line. While it is appreciated that performing many of these analyses in CLP cells is not feasible, the manuscript needs to acknowledge that EL4 cells are relatively unrelated to any of the cell types being considered and therefore there may be caveats for the interpretations based on data from them.

3) In Figure 4, it is unclear how the authors excluded contamination of the aLP by CHILP, which might skew the obtained results.

4) The authors refer to literature (24) for the impact of NFIL3 on NKP cells, but that study found only a small reduction of NKP cells in Nfil3-deficient mice. It would therefore be helpful if the authors could present data for NKP cells in Nfil3-deficient mice in their hands.

5) Figure 2 shows that Nfil3 deficiency results in a 90% reduction in total aLP cells. Nonetheless, T and B cell development is not affected. It would be very useful to show if the CXCR6+ aLP subset specifically and substantially depleted in Nfil3-deficient mice, or if depletion occurs in the CXCR6- aLP cells as well.

6) In Figure 5, the ILC profile from the Nfil3-deficient CD45.2+ hCD2- donor cells (without TOX overexpression) would be an important and interesting control.

7) Since the authors later show that only the CXCR6+ subset of aLP have the potential to give rise to all ILC lineages, it would be helpful if they could show the ILC transcription factor profile of these cells. Do they express any Id2? Do they express RORgt, since [31] showed that 40% of the CXCR6+ aLP in fetal liver were RORgt+?

8) The developmental rescue of NFIL3-/- precursors by TOX is of interest, but its quantitative significance would be best interpreted in comparison with the efficiency of rescue of these Nfil3-/- precursors with Nfil3 itself.

---

## [Author Response]

*1) Since Nfil3-deficient mice appear to have lymph nodes (e.g.,*
[47]*) and Peyer's patches albeit fewer and smaller (present manuscript), it is arguable that LTi cell function is largely preserved during the fetal period in them. Also, NKp46- ILC3 are only mildly reduced in Nfil3-deficient mice (*Figure 1*). On the other hand, the authors show convincing data for the deficiency of the CHILP compartment, shown to be a common progenitor for helper-like ILCs including LTi cells, in the absence of Nfil3. Is it possible, for example, that Nfil3 is required for the maintenance of ILC progenitors in adult mice but not as strictly required for ILC fate decisions in the fetus? There is also literature suggesting Nfil3-indepedent generation of some ILC subsets; salivary gland NK cells (*[4]*), cNK cells after MCMV infection (*[7]*), as also the mild reduction in NKP cells in Nfil3-deficient mice (*[24]*). The discussion would benefit from acknowledgment and attempted resolution of these issues*.

We thank the reviewers for raising this issue and offering insightful suggestions for amplifying our discussion. We have added an additional paragraph to the Discussion section that deals with these points.

*2) A fair proportion of the molecular connections between Nfil3 and Tox have been derived from studies with the EL4 lymphoma cell line. While it is appreciated that performing many of these analyses in CLP cells is not feasible, the manuscript needs to acknowledge that EL4 cells are relatively unrelated to any of the cell types being considered and therefore there may be caveats for the interpretations based on data from them*.

We agree that it is important to discuss this point and have now added the following sentences to the Results section:

“EL4 cells are derived from T lymphocytes, a CLP-derived lineage, and thus we cannot exclude the possibility that the regulatory relationship between *Nfil3* and *Tox* differs between T lymphocytes and CLPs. Nevertheless, our studies on CLPs and EL4 cells both support the idea that NFIL3 is an activator of *Tox* expression.”

*3) In*
Figure 4*, it is unclear how the authors excluded contamination of the aLP by CHILP, which might skew the obtained results*.

We cannot exclude contamination of the total αLP population by CHILP, and in fact, it is likely that CHILP cells are present in the αLP population. However, because CHILP do not give rise to cNK cells (Klose et al., *Cell* vol. 157, p. 340 [2014]), our data identify a distinct CXCR6^+^ αLP committed precursor that gives rise to all four ILC lineages including cNK cells (revised Figure 4). This precursor must therefore lie developmentally upstream of the CHILP. Additionally, in support of our *in vitro* findings, we now include data (new Figure 4) showing that the CXCR6^+^ αLP cells also give rise to all four ILC lineages (including cNK cells) *in vivo*.

*4) The authors refer to literature (*[24]*) for the impact of NFIL3 on NKP cells, but that study found only a small reduction of NKP cells in Nfil3-deficient mice. It would therefore be helpful if the authors could present data for NKP cells in Nfil3-deficient mice in their hands*.

Figure 1 of [24] actually shows a marked decline in NKP frequencies and absolute numbers in *Nfil3*^*-/-*^ mice. We note that the data are shown in log scale, compressing the visual differences. The percentages of both preNKP and rNKP cells among Lin- bone marrow cells declines by ∼8-10-fold in *Nfil3*^*-/-*^ mice. Likewise, there is an approximately 10-fold decline in the absolute numbers of both preNKP and rNKP cells in *Nfil3*^*-/-*^ mice. Therefore, the Male et al. paper strongly supports the idea that NFIL3 is essential for the differentiation of NKP cells.

*5)*
Figure 2
*shows that Nfil3 deficiency results in a 90% reduction in total aLP cells. Nonetheless, T and B cell development is not affected. It would be very useful to show if the CXCR6+ aLP subset specifically and substantially depleted in Nfil3-deficient mice, or if depletion occurs in the CXCR6- aLP cells as well*.

We have now included an analysis of CXCR6^+^ αLP cells in wild-type and *Nfil3*^*-/-*^ mice. The percentages of CXCR6^+^ cells among αLP are comparable between wild-type and *Nfil3*^*-/-*^ mice (Figure 4; Figure 4—figure supplement 1). As a result, the number of CXCR6^+^ αLP cells is markedly diminished in *Nfil3*^*-/-*^ mice (Figure 4) in parallel with the decrease in total αLP cells (Figure 2). These data support the idea that CXCR6^+^αLPs require NFIL3 for their development.

Since αLP cells do not give rise to B cells (Figure 3), it is not surprising that B cell development is not impaired in Nfil3-/- mice despite the marked αLP deficiency. Though αLP cells retain some T cell differentiation potential (Figure 3 and Figure 4), it is not clear to what extent αLP-derived T cells contribute to the total T cell pool in mice. There is evidence that LSK cells, which are present in normal numbers in Nfil3-/- mice, can enter the circulation and seed the thymus to initiate T cell development (Reviewed in Bhandoola and Sambandam, From stem cell to T cell: one route or many? Nat Rev Immunol. vol.6(2), p. 117-26[2006].) Thus, the marked αLP deficiency in *Nfil3*^*-/-*^ mice is consistent with the relatively normal T cell numbers in these mice.

*6) In*
Figure 5*, the ILC profile from the Nfil3-deficient CD45.2+ hCD2- donor cells (without TOX overexpression) would be an important and interesting control*.

The TOX encoding plasmid was delivered into LSK cells by retroviral transduction/infection. It is well-documented that retroviral infection of bone marrow progenitors can by itself impact their ability to differentiate (reviewed in Banerjee et al., Hematopoietic stem cells and retroviral infection. *Retrovirology* vol. 7, p. 8-17 [2010]). Therefore, comparison of hCD2^+^ and hCD2^-^ donor cells will not provide interpretable information about the impact of TOX expression on the differentiation potential of the cells. For this reason, we have restricted our analysis to a comparison between *Tox*-expressing hCD2^+^ donor cells and vector-only hCD2^+^ donor cells.

*7) Since the authors later show that only the CXCR6+ subset of aLP have the potential to give rise to all ILC lineages, it would be helpful if they could show the ILC transcription factor profile of these cells. Do they express any Id2? Do they express RORgt, since*
[31]
*showed that 40% of the CXCR6+ aLP in fetal liver were RORgt+?*

This is a great suggestion. However, the number of CXCR6^+^ αLP cells in adult mouse bone marrow is exceptionally small, perhaps due to the fact that these cells represent a transitional stage between αLP and CXCR6^+^ α4β7^-^ cells that lie downstream of αLP (31). The small number of cells makes such an analysis exceptionally challenging. As shown in revised Figure 4, we detect ∼5 CXCR6^+^ αLP cells per 3.3 million Lineage-negative bone marrow cells. As there are, on average, approximately 12 million Lineage-negative bone marrow cells per femur/tibia, this equates to ∼40 CXCR6^+^ αLP cells per femur/tibia. The rarity of these cells makes it extremely difficult to accurately assess transcription factor expression across the population in adult mouse bone marrow by flow cytometry. The analysis in [31] was performed on fetal liver CXCR6^+^ αLP cells, which account for roughly 0.15% of Linage-negative fetal liver cells and are thus much more abundant than in adult bone marrow. Finally, Possot et al. used quantitative RT-PCR to show that *Rorc* (encoding RORγt) expression is undetectable in CXCR6^+^ cells from adult bone marrow.

*8) The developmental rescue of NFIL3-/- precursors by TOX is of interest, but its quantitative significance would be best interpreted in comparison with the efficiency of rescue of these Nfil3-/- precursors with Nfil3 itself*.

This is a great suggestion and we have now included data showing the efficiency of developmental rescue of *Nfil3*^*-/-*^ precursors with NFIL3 in Figure 5. The results show that developmental rescue is comparable between TOX and NFIL3, with no statistically-significant differences between the two across the various ILC subsets. Although the average rescue efficiency for the ILC2 subset trended higher with NFIL3, the overall difference was not statistically significant.